# Porcine Astrovirus Infection in Brains of Pigs in Korea

**DOI:** 10.3390/v16091372

**Published:** 2024-08-28

**Authors:** Jun-Soo Park, Chang-Gi Jeong, Su-Beom Chae, Myeon-Sik Yang, Byungkwan Oh, Sook-Young Lee, Jae-Ku Oem

**Affiliations:** 1Laboratory of Veterinary Infectious Disease, College of Veterinary Medicine, Jeonbuk National University, Iksan 54596, Republic of Korea; spinyang@naver.com (J.-S.P.); jcg0102@gmail.com (C.-G.J.); cotnqja23@naver.com (S.-B.C.); 2Department of Companion and Laboratory Animal Science, Kongju National University, Yesan-eup, Yesan 32439, Republic of Korea; 111@kongju.ac.kr; 3Laboratory of Veterinary Pathology, Biosafety Research Institute, College of Veterinary Medicine, Jeonbuk National University, Iksan 54596, Republic of Korea; guroom2@gmail.com; 4Division of Life Sciences, Korea Polar Research Institute, Incheon 21990, Republic of Korea

**Keywords:** neuro-invasive, astrovirus, porcine, coinfection, encephalitis

## Abstract

Recently, neurological diseases associated with astroviruses (AstVs) have been reported in pigs, ruminants, minks, and humans. In 2017, neuro-invasive porcine astrovirus (Ni-PAstV) 3 was detected in the central nervous system (CNS) of pigs with encephalomyelitis in Hungary and the USA. In the process of diagnosing domestic pigs exhibiting neurological signs, histopathologic lesions of non-suppurative encephalomyelitis with meningitis, neuronal vacuolation, and gliosis were detected, and PAstV was identified using reverse transcriptase PCR in CNS samples of four pigs in three farms from August to September in 2020, South Korea. Subsequently, the ORF2 region was successfully acquired from three brain samples, facilitating subsequent analysis. Four genotypes of PAstV (PAstV1, 3, 4, and 5) were detected, and coinfection of PAstV with multiple genotypes was observed in brain samples. This is the first study to report Ni-PAstV infection in pigs in South Korea.

## 1. Introduction

Astroviruses (AstVs) are non-enveloped (~28–30 nm in diameter), single-stranded, positive-sense RNA (6.2–7.8 kb) viruses belonging to the family *Astroviridae* [1]. The genome of AstVs consists of three open reading frames (ORFs), ORF1a, ORF1b, and ORF2 [2], with two untranslated regions (UTRs) at 5′ and 3′ ends flanking the genome and polyadenylated at the 3′ end. This family comprises two genera: *Mamastrovirus* (MAstV) and *Avastrovirus* (AAstV). Members of the genus *Mamastrovirus* can infect mammalian hosts, including 19 species according to the 2023 classification of the International Committee on Taxonomy of Viruses (ICTV). On the other hand, members of the genus *Avastrovirus* that can infect avian hosts have three species. However, most newly discovered strains remain unclassified, including porcine astroviruses (PAstVs). In 2017, fourteen newly identified AstV strains were proposed as genotype species of the genus *Mamastrovirus*. PAstVs also belong to the genus *Mamastrovirus* and are divided into five distinct genotypes (PAstV1–5) based on sequence analysis of the open reading frame 2 (ORF2) region, which encodes the capsid protein [3,4,5]. It has been proposed that PAstV1 corresponds to MAstV-3, PAstV2 to MAstV-31 and MAstV-32, PAstV3 to MAstV-22, PAstV4 to MAstV-26 and MAstV-27, and PAstV5 to MAstV-24 [6].

AstV was first identified in 1975 by electron microscopy of feces collected from children with diarrhea [7]. Since then, astroviral enteric infections causing gastroenteritis in infants and children have been reported worldwide [8,9,10]. Similar to humans, PAstV is commonly considered a cause of gastroenteritis in swine [3,11]. Recently, AstVs have been suggested as novel viral pathogens outside the gastrointestinal (GI) tract [12,13]. After the first discovery of the association between AstV infection and human gastroenteritis, subsequent findings included duck AstV hepatitis in 1984 [14], avian nephritis virus nephritis in 2000 [15], and AstV VA1/HMO clade encephalitis in 2010 [16,17]. Moreover, AstV was suspected to be a cause of a respiratory disease [13], AstV MLB2 febrile respiratory disease in 2012 [17], and PAstV4 of pigs in 2016 [18]. In 2010, both neurological signs and central nervous system (CNS) infections of AstV were reported in humans and minks [16,19]. Following the first report of neuro-invasive astroviruses (Ni-AstVs), more evidence of Ni-AstVs strains has been reported in cattle [20], sheep [21], pigs [22,23], musk oxen [24], and alpaca [25]. In 2021, an infection of bovine AstV causing nonsuppurative meningoencephalitis was confirmed in cattle in South Korea [26]. Furthermore, in 2023, AstV was detected in the brains of raccoon dogs of South Korea [27]. Based on current knowledge, AstVs causing neurological symptoms in mammals mainly belong to the VA/HMO clade of *Mamastrovirus* genogroup II. Human AstVs of MLB clade (*Mamastrovirus* genogroup I) with neurological symptoms have also been reported [28].

AstVs demonstrate high genetic diversity across various hosts, with frequent reports of high diversity even within a single host [29]. This tendency is particularly pronounced in densely populated or domesticated hosts such as pigs, humans, bats, and bovines [6]. A study conducted in the USA reported a 13.9% rate of co-infection of multiple PAstV genotypes in pigs [3]. Diverse genotypes of co-infection were thought to increase the opportunity for genetic recombination [3]. The original PAstV has been known as an enteric virus. Investigations have mainly been conducted using fecal samples. However, a study reported in 2014 raised concerns about the association between encephalitis and PAstV2 and 5 detected in the brain [30]. Several evidence supported the association of PAstV with cases of extraintestinal infections, including CNS infections of pigs [28,30,31,32,33]. In 2017, neuro-invasive PAstV (Ni-PastV) 3 was detected in the CNS of pigs with encephalomyelitis in Hungary and the USA [22,23]. Infected pigs exhibited clinical signs of astasia and knuckling, with a fatality rate of 75–100% [22]. Ni-PAstV3 also belongs to the genogroup II of *Mamastrovirus* in phylogenetic analysis. Since the discovery of Ni-PAstV3 in pigs in 2017, examinations using central nervous system samples have been actively conducted [33,34,35,36,37]. In South Korea, PAstV2 and PAstV4 were reported in both domestic pigs and wild boars [38,39,40]. However, previous studies in Korea have only analyzed fecal samples without considering CNS infection or disease signs. In this study, the presence of Ni-PAstV in South Korea was investigated using samples collected from domestic pigs showing neurological signs.

## 2. Materials and Methods

### 2.1. Sample Collection

Carcasses of five pigs with neurological signs (astasia, knuckling, and pedaling with hind limb stretching) from three farms were submitted to Jeonbuk National University Veterinary Diagnostic Center (JBNU-VDC) from August to September 2020. The pigs ranged in age from 4 to 8 weeks, including both suckling and weaned piglets, with symptoms primarily appearing among littermates. The main symptom observed was the inability to stand; affected piglets that had fallen over would repeatedly exhibit paddling movements but were unable to rise easily. Case numbers 20-0983 and 20-0932-1 (8-week-old) and 20-1295 (4-week-old) pigs were from farms in Jeongeup, Jeollabuk-do, Korea. The pig with case number 20-1006-1 and 20-1006-2 (4-week-old) was from a farm located in Hamyang, Gyeongsangnam-do, Korea. Each case exhibited a sporadic outbreak pattern within the farm, with several instances of mortality observed. The specimens were transported immediately after the animals were slaughtered for prompt examination. Autopsy was conducted for each pig on different days. Tissue samples from various parts of the central nervous system, including cerebrum and cerebellum and spinal cord, were collected from each pig. These tissues were stored at −80 °C until further study or fixed in 10% neutral buffered formaldehyde to prepare formalin-fixed paraffin-embedded (FFPE) blocks.

### 2.2. RNA Extraction and cDNA Synthesis

Tissue samples were washed twice with 5 mL of phosphate-buffered saline and homogenized in 1 mL of phosphate-buffered saline using TissueLyser II (Qiagen, Hilden, Germany). Homogenized samples were then centrifuged at 16,000× *g* for 10 min. RNA extraction and cDNA synthesis were performed. Briefly, 255 µL of homogenate supernatant was mixed with 30 μL of 10× DNase buffer and 150 U DNAse I (Roche, Basel, Switzerland) to have a total volume of 300 µL. The mixture was then incubated at 37 °C for 2 h. Next, 500 µL of TRIzol Reagent (Ambion, Austin, TX, USA) was added to the sample mixture and incubated at room temperature for 5 min. Subsequently, 200 µL of chloroform was added to the mixture and incubated at room temperature for 3 min. The sample mixture was centrifuged at 16,000× *g* for 10 min at 4 °C. RNA was extracted from the RNA contained aqueous phase using an RNeasy Mini Kit (Qiagen, Hilden, Germany), and cDNA was synthesized using an AccuPower RocketScript RT PreMix (Bioneer, Daejeon, Republic of Korea) and a universal reverse adaptor primer (AP) with 17 dT at the 3′ end.

### 2.3. PCR Screening and Sequencing

For detecting PAstV, partial RdRp sequence of AstV was amplified using published primer sets [41,42] with hemi-nested PCR (Table 1). For the first-round PCR, AstV Pol F1, AstV Pol F2, and AstV Pol R1 were used. For the second-round PCR, AstV Pol F3, AstV Pol F4, and AstV Pol R1 were used. PCR was performed under the following conditions: 94 °C for 1 min; 40 cycles of 94 °C for 30 s, 50 °C for 30 s, and 68 °C for 30 s; and a final extension at 68 °C for 5 min. ORF2 sequences were amplified using PAstV1-5 ORF2 F as the forward primer and the conserved stem–loop II-like motif (s2m) at the 3′ end of PAstV1, 3, and 5 [3] with AstV s2m R as the reverse primer, while PAstV2 and 4 were amplified using the 3′ RACE method with AP. PCR products in different length were separately purified and sequenced using next-generation sequencing-based technology on the illumina MiSeq platform, barcode-tagged sequencing (BTSeq^TM^ Services; Celemics, Seoul, Republic of Korea). All ORF2 sequences were deposited in NCBI database by accession no. OP643769.1 to OP643782.1.

### 2.4. Sequence Analysis

Phylogenetic analyses were performed based on ORF2 amino acid sequences. These sequences were edited using BioEdit v. 7.2.5 [43] and aligned using Clustal Omega [44]. Phylogenetic trees were constructed using the neighbor-joining method and the p-distance model with 1000 bootstrap replicates with the MEGA X program [45]. Nucleotide and amino acid sequence identities were calculated using BioEdit using the sequence identity matrix method (p-distance).

### 2.5. Other Laboratory Diagnostics

Samples were tested for the presence of eight viral pathogens and six bacterial pathogens, including those causing neurological diseases according to diagnostic manuals of the JBNU-VDC (porcine circovirus type 2 virus (PCV2) and porcine reproductive and respiratory syndrome virus (PRRSV); Prime-Q PRRSV/PCV2 Detection Kit (Genet Bio, Daejeon, Republic of Korea), porcine epidemic diarrhea virus (PEDV) and transmissible gastroenteritis virus (TGEV); Prime-Q PEDV/TGEV Detection Kit (Genet Bio, Daejeon, Republic of Korea), Aujeszky’s disease virus (ADV); PCR [46], classical swine fever virus (CSFV); RT-PCR [47], Japanese encephalitis virus (JEV); RT-PCR [48], swine influenza virus (SIV); RT-PCR [49], (*Brachyspira hyodysenteriae* (swine dysentery); PCR [50,51], *Escherichia coli* (*E. coli*); PCR [52,53], *Haemophilus parasuis* (Glasser’s disease); PCR [54], *Lawsonia intracellularis*; PCR [55], *Listeria* spp.; PCR [56], and *Salmonella* spp.; real-time PCR [57]).

Bacterial cultivation was conducted to identify additional infectious pathogens. Subdural swab samples from the brains were cultured on MacConkey agar (MB-M1028; MB-cell, Gyeonggi-do, Republic of Korea) and blood agar plates (AM601-02; Asan Pharm, Gyeonggi-do, Republic of Korea). Each colony was identified by sequencing the 16S rRNA gene.

### 2.6. Histopathology

FFPE blocks were sectioned to 4 μm in thickness and used for hematoxylin and eosin (H&E) staining. H&E staining was conducted using a standard protocol [58]. A histopathological examination was conducted at the JBNU-VDC.

## 3. Results and Discussion

Partial RdRp sequences of PAstV were successfully amplified using a heminested PCR from brain tissue samples of four pigs, except the 20-0983-02 pig. The possibility of co-infection of multiple genotypes of PAstV was suggested by partial RdRp sequence analysis. ORF2 sequence analysis from three pigs identified the presence of PAstV genotypes 1, 3, 4, and 5 with multiple genotypes coexisting within 20-1006-2 and 20-1295 (Table 2). In South Korea, the prevalence of PAstV in domestic pigs was reported to be 20.1% and 9.2% in wild boars [38,40]. PAstV4 was dominant (94.6%) in swine, with a few PAstV2 present (5.4%) [38,39,40]. The detection of PAstV1, 3, and 5 genotypes represents the first detection in South Korea in this study. Six sequences including complete capsid protein encoded ORF2 sequence with lengths ranging from 2247 to 2937 nucleotides and one sequence (PAstV4 KOR/1006-2/2020 Brain (OP643774)) with partial ORF2 sequence with a length of 3104 nucleotides were obtained. A phylogenetic tree (Figure 1A) was constructed with PAstV genotypes 1–5, *Avastroviruses* 1–3, and reported Ni-AstV and PAstV using ORF2 amino acid sequences. Seven sequences obtained from brain samples of Korean pigs were closely related to known PAstV genotypes 1, 3, 4 and 5. PAstV 1, 4, and 5 were classified into the *Mmastrovirus* genogroup I, along with the MLB clade. PAstV3, on the other hand, was classified into genogroup II with the VA/HMO clade, which included the majority of Ni-AstV strains. 

In molecular analysis, the obtained sequences showed high genetic diversity. Phylogenetic analysis and sequence comparison of the ORF2 sequence of PAstV1 were conducted using seven reference sequences (Figure 1B). Although PAstV1 represents the earliest genotype discovered among PastVs, reported sequence data are limited. The highest amino acid sequence identity with the PAstV1 KOR/1006-2/2020 brain was 80.6% (KF787112.3; China), while the lowest identity recorded was 70.6% (MW504546.1; USA) (see Appendix A). A comparison with Porcine/South Africa/BSF2/2021 (OM105035.1) sequence in the conserved acidic amino acid-rich region located at the C-terminal part of the ORF2 sequence revealed that the PAstV1 KOR/1006-2/2020 brain contained deletions of 10 amino acids at true positions 645–646 (see Appendix A).

In comparison with 69 reference sequences, the PAstV4 KOR/1006-2/2020 brain amino acid sequence exhibited the highest identity of 70.6% with Porcine/USA/P2011-1/2011_(JX684071.1) and the lowest identity of 38.2% with Porcine/China/HLJ01_C13/2017_(MK378532.1) (see Appendix A). Phylogenetic analysis showed that PAstV4 tended to be divided into two distinct groups, with KOR/PAstV4 included in group A along with 15 reference sequences (Figure 1D). Amino acid sequence comparisons revealed the presence of the MVGIDSTKPFFL motif at the 418–429 amino acid position within the extended conserved region of the N-terminal portion of the ORF2 sequence of the PAstV4 KOR/1006-2/2020 brain (see Appendix A). Similar motifs were observed in group A.

Phylogenetic analysis of PAstV5 was conducted with four Korean sequences and 42 reference sequences (Figure 1E). The highest amino acid identity in each sequence was observed as follows: 97.0% between PAstV5 (MW504545.1; USA) and PAstV5 KOR/0983/2020 brain1, 98.9% between PAstV5 (KP747574.1 China) and PAstV5 KOR/0983/2020 brain2, 98.5% between PAstV5 (KP747574.1 China) and PAstV5 KOR/1006-2/2020 brain, and 89.9% between PAstV5 (MW504545.1; USA) and PAstV5 KOR/1295/2020 brain (see Appendix A). Two different PAstV5 sequences were obtained from KOR/0983/2020. They were separated into two distinct groups in the phylogenetic analysis, with PAstV5 sequences detected in 20-1006-2 and 20-1295, each. They shared 66.5% of nucleotide identities and 73.6% of amino acid identities. Meanwhile, PAstV5 KOR/0983/2020 brain2 with the PAstV5 KOR/1006-2/2020 brain shared 93.7% of nucleotide identities and 99.0% of amino acid identities. In addition, PAstV5 KOR/0983/2020 brain1 with the PAstV5 KOR/1295/2020 brain shared 83.8% of nucleotide identities and 89.9% of amino acid identities.

Phylogenetic analysis of the PAstV3 KOR/1295/2020 brain with 40 reference sequences showed an interesting feature. A distinct cluster comprising the PAstV3 KOR/1295/2020 brain and two reference sequences (KY933399.1; Uganda, LC201598.1; Japan) was observed, contrasting with the group containing previously reported Ni-PAstV3 strains (Figure 1C). Amino acid sequence comparisons of the PAstV3 KOR/1295/2020 brain showed the highest amino acid identities of 91.8% (KY933399.1; Uganda), whereas it showed amino acid identities ranging from to 59.2% to 60.5% with Ni-PAstV3 strains reported in the USA and Hungary in 2017, respectively (KY940545.1; USA and KY073230.1–KY073232.1; Hungary) (see Appendix A). The majority of Ni-AstVs in the HMO clade are known to possess the Q(I/L)QxR(F/Y) motif [59]. The function of this motif remains unknown; however, interestingly, a similar motif (EIQRRF) was also identified in the PAstV3 KOR/1295/2020 brain (see Appendix A). This sequence was even more similar to the Ni-AstV sequence reported in minks (AIQRRF) (GU985458; Sweden) than to the reported Ni-PAstV3 sequence (QIQQRF). Remarkably, it exhibited a closer resemblance to the sequence detected in the brain tissue of Korean raccoon dogs (VIQRRF) [6,19,23,27]. Cases of interspecies transmission of AstVs have been reported [12,13], and traces of interspecies transmission associated with PastVs have also been identified [60]. The Q(I/L)QxR(F/Y) motif of PAstV3 identified in this experiment, although found in pigs, may exhibit higher identity sharing with AstVs from different species (raccoon dogs and minks) than with previously reported PAstV3. These observations may also signal instances of interspecies transmission that have occurred with PAstVs. 

The low sequence identities with <95% identity at the nucleotide sequence level could lead to significant serological differences [61,62]. The amino acid sequence identity of PAstV5 below 95% could also induce significant serological differences. In ICTV, within each *Mamastrovirus* genogroup, capsid protein amino acid genetic distances (p-dist) between genotypes range from 0.338 to 0.783 [63]. Interestingly, Korea PAstV4, 5 and 3 with Ni-PAstV sequences were divided into at least two small clades (Figure 1A–E). Observed differences in sequence similarity in amino acid and the tendency of grouping within the genotype in phylogenetic analysis suggest a potential for further subtyping within the genotype of PAstV.

The JBNU-VDC performed PCR for viral and bacterial pathogens and bacterial cultivation to identify the cause of neurological symptoms and encephalitis (see Appendix A). The lung sample from the 20-0983 pig was PCR-positive for PRRSV. *Streptococcus suis* was isolated from brain tissues. The lung sample from the 20-1295 pig was PCR-positive for PRRSV. *Streptococcus suis* and *Clostridium perfringens* type A were isolated from the brain and intestine, respectively. In the case of specimen 20-1006-2, major tests for the causative agent of encephalitis yielded negative results.

When Ni-AstV infects CNS regions such as the brainstem, cerebellar/cerebral cortex, hippocampus, and spinal cord, it appears to infect various types of neurons, including Purkinje cells, interneurons, and CA pyramidal neurons, as well as glial cells such as astrocytes. This infection is associated with pathological conditions such as neuronal degeneration, necrosis, neuronophagia, and gliosis [59]. Various CNS inflammations such as encephalitis, meningitis, and meningoencephalomyelitis have been observed depending on the animal species infected with Ni-AstV [16,22,23,25,59,62,63,64]. CNS histopathology detected inflammation in both the brain and spinal cord (Figure 2). Meningitis was detected in the cerebrum and cerebellum of 20-0983 and 20-1295 (Figure 2A,E; 20-0983, Figure 2O,P; 20-1295). Non-suppurative inflammation, encephalitis with perivascular cuffing (Figure 2C,G; 20-0983), and neuronal necrosis of Purkinje cells in the brain with degeneration were detected (Figure 2I; 20-0983). The spinal cord showed myelitis with perivascular cuffing (Figure 2K; 20-0983) and neuronal degeneration with necrosis (Figure 2M; 20-0983).

In pigs, the predominant CNS pathology found in Ni-PAstV3 infection was encephalomyelitis. However, previous studies have shown that in addition to encephalomyelitis, meningitis is also seen in Ni-PAstV3 infection [36]. On the other hand, *Streptococcus suis* is also characterized by meningitis as the main manifestation [65]. In 20-0983 and 20-1295, PAstV and *Streptococcus suis* were detected together in the brain. Meningitis was also observed. Thus, the cause of morbidity and mortality in pigs could not be clearly linked to PAstV infection. However, in 20-0983, histopathological examination of the spinal cord showed perivascular cuffing and neuronal degeneration, suggesting viral myelitis. In addition, PCR detected PAstV. Previous studies have detected PAstV5 in the brains of pigs [33]. Based on these results, the PastV5 detected in this study is likely to be a Ni-PAstV strain. Based on the results of this study and previous studies that have detected PAstV2 and PAstV5 in the brain [30], it is likely that other PAstVs, in addition to Ni-PAstV3, have the potential to cause central nervous system infections. However, the relationship between the detected PAstV in the brain and clinical symptoms could not be clearly established. Additional analysis of full genome sequence and type-specific ISH and/or in vivo experimental infection may help us further elucidate the pan-infection of PAstVs in the CNS.

## 4. Conclusions

In this study, PAstV genotype 1, 3, 4, and 5 were detected in three brain samples with the coexistence of multiple genotypes. The results of the histopathological examination confirmed signs of encephalitis, myelitis, and meningitis. These findings suggest that, in addition to PAstV3, other genotypes of PAstV can also infect the CNS of pigs, although further experiments are needed to confirm the neurotropism of these viruses. From a different perspective, this study is significant in that it represents the first detection of PAstV1, PAstV3, and PAstV5 genotypes, which have not been previously reported in South Korea.

## Figures and Tables

**Figure 1 viruses-16-01372-f001:**
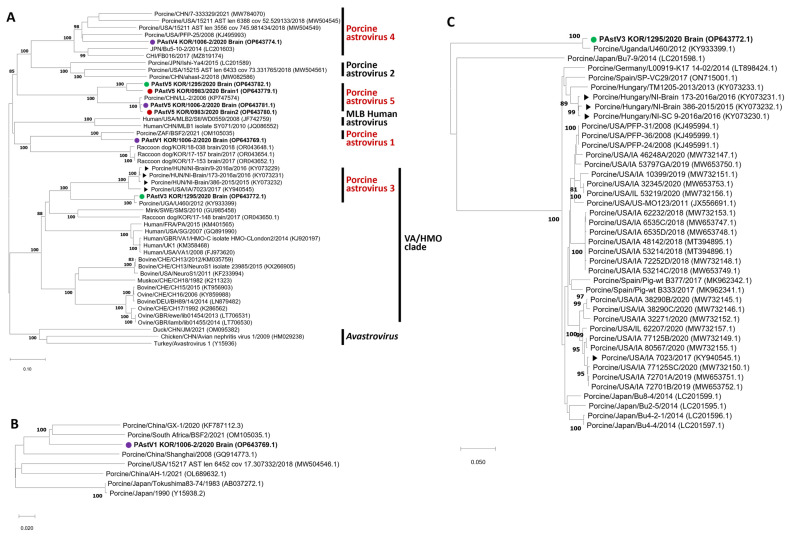
Phylogenetic tree of ORF2 amino acid sequences of PAstVs and Ni-AstVs. The tree was constructed using the neighbor-joining and p-distance methods with 1000 bootstrap replications. Sequences of Korean pig samples are marked as follows. (

) KOR/0983/2020, (

) KOR/1006-2/2020 and (

) KOR/1295/2020, while Ni-PAstV3 sequences of USA/IA/7023/2017 and NI-Brain/HUN strains are marked with (▶). (**A**) PAstVs with Ni-AstVs and AastVs; (**B**) PAstV1; (**C**) PAstV3; (**D**) PAstV4; (**E**) PAstV5.

**Figure 2 viruses-16-01372-f002:**
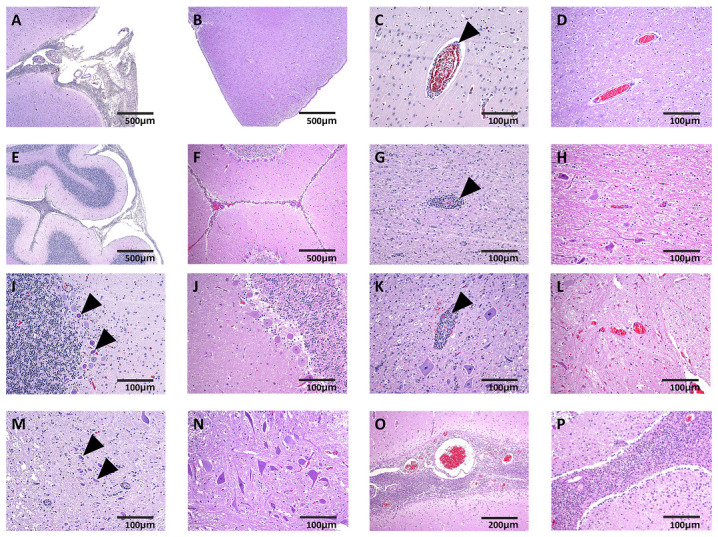
Tissue sections of brain (**A**–**J**,**O**,**P**) and spinal cord (**K**–**N**) stained with hematoxylin and eosin (**A**,**C**,**E**,**G**,**I**,**K**,**M**; 20-0983. **O**,**P**; 20-1295. **B**,**D**,**F**,**H**,**J**,**L**,**N**; negative control (NC)). (**A**,**O**) Cerebrum with meningitis and (**B**) NC; (**C**) cerebrum with perivascular cuffing (black arrowhead) and (**D**) NC; (**E**,**P**) cerebellum with meningitis and (**F**) NC; (**G**) cerebellum with perivascular cuffing (black arrowhead) and (**H**) NC; (**I**) Purkinje cell necrosis and degeneration of cerebellum (black arrowhead) and (**J**) NC; (**K**) spinal cord with perivascular cuffing (black arrowhead) and (**L**) NC; (**M**) necrosis and degeneration of neurons of spinal cord (black arrowhead) and (**N**) NC. Scale bar, 500 μm (**A**,**B**,**E**,**F**), 200 μm (**N**), and 100 μm (**C**,**D**,**G**–**N**,**P**).

**Table 1 viruses-16-01372-t001:** Primers used in PCR.

Primer		Sequence (5′→3′)	Target Region	Position	Position Reference (GenBank No.)
AstV Pol [41]		F1: GARTTYGATTGGRCKCGKTAYGA	ORF1b (RdRp)	3498–3916	KF787112
		F2: GARTTYGATTGGRCKAGGTAYGA			
		F3: CGKTAYGATGGKACKATHCC		3513–3916	
		F4: AGGTAYGATGGKACKATHCC			
		R: GGYTTKACCCACATNCCRAA			
ORF2	ORF2 F	F: CTSYATGGGAAACTCCT	ORF2	4065–6597	KF787112
	s2m	R: CCCTCGATCCTACTCGG			
	AP-dT17	R: GGCCACGCGTCGACTAGTAC-Oligo(dT)17			
	AP	R: GCCACGCGTCGACTAGTAC			

**Table 2 viruses-16-01372-t002:** Identified PAstV genotypes from PCR results and sequence analysis.

Samples			Target		Sequence (Accession No.)	Length (CDS)
No.	Location	Organ	ORF1b *(RdRp)*	ORF2 (Capsid)		
20-0983	Jeollabuk-do, jeongeup †	Brain	+	5	PAstV5 KOR/0983/2020 Brain1 (OP643779)	2247 (2226)
					PAstV5 KOR/0983/2020 Brain2 (OP643780)	2265 (2208)
20-1006-2	Gyeongsangnam-do, hamyang	Brain	+	1, 4, 5	PAstV1 KOR/1006-2/2020 Brain (OP643769)	2496 (2313)
					PAstV4 KOR/1006-2/2020 Brain (OP643774)	3104 (2465)
					PAstV5 KOR/1006-2/2020 Brain (OP643781)	2937 (2208)
20-1295	Jeollabuk-do, jeongeup †	Brain	+	3, 5	PAstV3_KOR/1295/2020 Brain (OP643772)	2788 (2298)
					PAstV5_KOR/1295/2020 Brain (OP643782)	2437 (2238)

†; The two samples were obtained from two separate farms in the same region. +; PCR positive. CDS; coding region.

## Data Availability

The data presented in this study are openly available in NCBI database by accession no. OP643769.1, OP643772.1, OP643774.1 and OP643779.1-OP643782.1.

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
