# Peer review of "Porcine Astrovirus Infection in Brains of Pigs in Korea"

_viruses, 2024, doi:10.3390/v16091372_

Round 1

Reviewer 1 Report

Comments and Suggestions for Authors

This is a brief report showing the presence of porcine astrovirus in pigs with neurological symptoms in Korea. Although it is an interesting report not all the data needed to undertand the article is shown, and some experiments that were performed and mention, are shown, neither mention in methodology.

Although I imagine a short report should have 2-3 figures, it would be very important that the authors show the PCR amplification, specially because they are using nested PCR. They should also show the results and explain the methodology for the detection of the other infectious agents they analyzed.

They should also mention either in methodology or the results that they deposit the sequences of the ORF2 in NCBI, they showed the accession number, but one must assume they were reported here.

Since the authors are making comments in the results and discussion section about the sequences, indication that fragments are missing, they should include, maybe as supplementary, the alignments between the cloned sequences that were used for the analysis, labeling the main differences/similarities.

In lines 205 the authors discuss about motifs that are variable between reference sequences and the sequences found, they should explain better the importance of this motif for the function of the protein encoded in ORF2.

The legend of figure 1 is not clear, two different viruses have the same symbol, and the colors coding in not named or explained in the legend.  The figure itself is difficult to read, would it be possible to separate it in two or tree?

Figure 2, occupies 2 different pages (maybe can be divided into two figures, according to the area shown), although to what corresponds each panel is indicated in the figure, the tissue area that is showing the meningitis, perivascular cuffing and all the pathological markers indicated in the legends, should be clearly labeled in each photohgraph. More importantly, a normal tissue should be included for comparison.

As presented now this article is not suitable for publication, since not all data mentioned and needed to support the conclusions is shown.

Comments on the Quality of English Language

Ther are some grammar things that need to be improved

Author Response

Thank you for giving us the opportunity to strengthen our manuscript with your valuable comments and queries. We have attached the revision response file below. We have worked hard to incorporate your feedback and hope that these revisions persuade you to accept our submission.

Reviewer 2 Report

Comments and Suggestions for Authors

The study of Park and co-workers describes the detection and phylogenetic/sequence analyses of multiple different porcine astrovirus-related sequences from “brain” samples of diseased pigs with neurological symptoms. The findings are overall very promising, but beside several review rounds the manuscript is still preliminary and still contains multiple flaws which needs to be addressed properly. I suggest the complete revision of the manuscript (see the detailed points below).

Main points:

The main problem with the study is that there is no convincing evidence which could support the neurotropic nature of the study PAstVs identified from the “brain” samples. These viruses as contaminants can be originated from the intestine during autopsy which can be amplified by applied RT-PCR methods. PAstV-type specific in situ RNA hybridization (like RNAScope) techniques on available FFPE samples or at least quantitative assays (e.g.: PAstV-type specific RT-qPCR) should be conducted on different CNS samples to prove/support the neurothropic potential of these PAstVs. The conclusions (multiple, even co-infecting PAstVs can cause neuroinfection in swine) are not supported by the presented results.

Beside of these, no adequately detailed description has been found of the disease and investigated samples. The age, holding conditions and other relevant information of the investigated pigs should be included. Were all the diseased pigs from the different farms showed the same symptoms? Were the pigs died because of the disease? How many pigs were affected? Was there a local epidemic?

The authors mention that multiple samples were collected from five carcasses (lanes 86 - 87, 92-93 “Brain tissue samples with cerebrum and cerebellum and spinal cord were collected from each pig.”) while only “brain” samples were mentioned throughout the whole manuscript and all PAstV strain names of the study as well as in Table 2. This sentence should be re-phrased as “tissue samples from various parts of the central nervous system including cerebrum and cerebellum and spinal cord were collected from each pig.”

Furthermore, according to the GenBank records, some of the study sequences (OP643770.1; OP643771.1, OP643773.1, OP643775.1, OP643776.1, OP643777.1, OP643778.1) indicated in lines 299-300 were originated from “intestine”. Were enteric samples also tested? These sequences are not included in the manuscript but should be analyzed and discussed in the text.

There is no information about the RT-PCR positivity and no sequence data either of the specific “brain” samples (cerebrum, cerebellum) or the spinal cord (except a short sentence in line 268-270: “In addition, PCR detected PAstV”). There is no information about the epidemic or sporadic presence of the disease associated with PAstV infections in the investigated farms (no morbidity/mortality data). The authors mention 5 carcasses (line 86), but details of animals were given in three cases 20-0983, 20-1295, 20-1006-2 (lines 88-90).

The description of how the sequence data were acquired are nearly completely missing. There is only a single sentence in lines 120 - 121 which indicate the method. Is it an NGS-based technique? There is also no reference or description about this “barcode-tagged sequencing” provided. The description of the sequencing method should be provided in a more detailed way.

Minor points:

Line 30: The term “They” should be re-placed with “the genome of AstVs”

Line 41: The term “based on the open reading frame 2” should be re-placed with “based on the sequence analyses of open reading frame 2”

The introduction section needs to be clarified. Should contain background information about the findings (e.g. lines 43-46 can be omitted, REF.[9] is not describes the the epidemiology of AstV but the epitopes) and should be clarified.  Furthermore, as far as I know not AstV but noroviruses are the second most common cause of viral gastroenteritis in children, after rotavirus (see: Bon, F., Fascia, P., Dauvergne, M., Tenenbaum, D., Planson, H., Petion, A. M., ... & Kohli, E. (1999). Prevalence of group A rotavirus, human calicivirus, astrovirus, and adenovirus type 40 and 41 infections among children with acute gastroenteritis in Dijon, France. Journal of clinical microbiology, 37(9), 3055-3058.) Please correct it or insert propriate reference(s) supporting the original statement.

Line 41: The novel nomenclature of PoAstVs should be mentioned at least here (PoAstV-1 was classified into genotype species MAstV-3; some of the viruses of PoAstV-2 lineage were classified into MAstV-31 and MAstV-32; PoAstV-3 was classified into MAstV-22, some of the PoAstV-4 viruses were classified into MAstV-26 and MAstV-27, while PoAstV-5 was classified into MAstV-24) See Chapter 7.2 in Boujon, C. L., Koch, M. C., & Seuberlich, T. (2017). The expanding field of mammalian astroviruses: opportunities and challenges in clinical virology. Advances in virus research, 99, 109-137. and Donato, C., & Vijaykrishna, D. (2017). The broad host range and genetic diversity of mammalian and avian astroviruses. Viruses, 9(5), 102.

Lines 46-48: Reference [2] is about human astroviruses not animal ones, please find more relevant reference(s) supporting the statement. Most of the animal AstVs are not clearly associated with gastroenteritis, see the review of Boujon et al. (Boujon, C. L., Koch, M. C., & Seuberlich, T. (2017). The expanding field of mammalian astroviruses: opportunities and challenges in clinical virology. Advances in virus research, 99, 109-137.)

The manuscript contains multiple scientific mistakes e.g.: PAstVs have no “genes” only genome parts/regions, there is no “amplification” during cDNA synthesis (in line 108), AstV typing is based on the sequence analysis of ORF2 (capsid) and not the RdRP (lines 147-148). Interspecies transmission of AstVs its rare and not “well documented” lines: 218-219) and several confusing sentences (e.g. lines 222-223).

Lines 173-204: This section is hard to read. Sequence comparison data should be presented as a Table.

Lines: 177, 178, 183, 184 and throughout the manuscript: Sequence similarity values should be replaced with identity values! These terms (similarity/identity) are not interchangeable.

Figure 1: Please include description about the colored circles.

Figure 2: The signs of “Meningitis“ and “Perivascular cuffing“ and “Necrosis with degeneration of neurons and myelitis“ should be indicated in the figures. Legend mentions “black arrowhead” which is missing from the figure!

Author Response

Thank you for giving us the opportunity to strengthen our manuscript with your valuable comments and queries. We have attached the revision response file below, and we have highlighted the major changes in the manuscript in yellow. We have worked hard to incorporate your feedback and hope that these revisions persuade you to accept our submission.

Reviewer 3 Report

Comments and Suggestions for Authors

Reviewer: While reviewing the manuscript “Porcine astrovirus infection in brains of pigs in Korea” I am of the opinion that the manuscript bears high quality original work with high importance to researchers in the field on the field of pathogens and pathogen-host interactions. However, I would like to mention several points that I feel should be addressed before I could recommend this manuscript for publication.

Comment 1: Has Ni-PAstV infection been previously reported in South Korea, or is this the first study documenting it in the country?

Comment 2: The clinical symptoms of affected pigs are less well described, and whether more detailed clinical data can be added to help understand the manifestations of the disease

Comment 3: Subsequent animal infection experiments can be carried out to verify whether these PAstV strains can cause nervous system symptoms to meet Koch's law.

Comment 4: In this study, only ORF2 gene sequence was obtained, and whole genome sequencing could provide more comprehensive viral genetic information.

Comment 5: Page 7, Line 219-220, “Traces of interspecies transmission associated with PastVs have also been 219 identified (Ulloa and Gutiérrez 2010).” Reference format used incorrectly, please unify format.

Comments on the Quality of English Language

Author Response

(The authors gave the same response as above.)

Round 2

Reviewer 1 Report

Comments and Suggestions for Authors

This article has greatly improved, I appreciate that the authors have included my recomendations and comments in this new version. 

I have only one concern taht was not adressed and is that the data for detection of other poathogens is not inlcuded in this version, not even as a supplementary figure. 

Minor comments are:

-Centrifugation speed shopuld be indicated as g not as rpm. 

-There is indistinct use of minute and min, please correct, it should be min. 

-line 174. Obteained seven sequences ===>Seven sequences obtained

-line 192. of (Figure 1B)===> eliminate of

-Fig sup3. The color of the rectangle marking Q(I/L)QxR(F/Y) motif is really hard to see, either chanche the colro of the rectangle or underline the are of the motif, or do both as in Fig sup2 for labelin groupA

Author Response

We sincerely appreciate the valuable feedback provided during the major revision process and are grateful for the continued guidance offered in this minor revision. We have attached the revision response file below. We hope the changes made align with your expectations.

Reviewer 2 Report

Comments and Suggestions for Authors

The authors suggesting that multiple types of PAstVs could cause (neuro)infection in swine but based on the results presented in this manuscript the neurotropism of these viruses are still not confirmed. The presence of a virus in the brain tissue does not necessarily means that the virus can infect brain cells and successfully propagated in it. The detected PAstVs could be originated from the intestine as contaminants or could be presented in the brain because of a viraemia/disseminated infection. The neurotropism of PAstV-3 was confirmed by ISH, and RT-qPCR (highest viral load in the CNS) which confirmation is still lacking in this study. If the authors can confirm the infection of PAstVs in brain tissue by RdRp-based ISH (as stated in the responses), then these results should be included into the manuscript otherwise the manuscript should be re-phrased to include this uncertainty. Like in the conclusion section the term “…although further experiments are needed to confirm the neurotropism of these viruses.” Should be included after the “These findings suggest that, in addition to PAstV3, other genotypes of PAstV can also infect the CNS of pigs” sentence.

2.1. Sample collection section: The information about the carcasses being omitted from the study should be included in the manuscript.

Data Availability Statement section: If the authors did not analyze PAstV sequences from intestine, please remove these GenBank accession numbers from the Data Availability Statement section.

Author Response

Thank you for your thorough review and thoughtful suggestions. We have attached the revision response file below. We hope the changes made align with your expectations.
